# EMAeHealth, a digital tool for the self-management of women's health needs during pregnancy, childbirth and the puerperium: protocol for a hybrid effectiveness-implementation study

Maite Espinosa Cifuentes  ,[1,2] Isabel Artieta-Pinedo,[2,3,4] Carmen Paz-Pascual,[2,5,6] Paola Bully-Garay,[7] Arturo García-Alvarez,[1,2] ema-Q Group[8]

For numbered affiliations see end of article.

**Correspondence to**
Dr Maite Espinosa Cifuentes;
maite.espinosacifuentes@
osakidetza.eus

## ABSTRACT

**Introduction** EHealth can help health service users take a more active role in decision-making and help health professionals guide the patient in this process. A digital tool has been designed to support maternal education (ME), and it is organised into four areas: (1) information, (2) communication, (3) health self-management and (4) clinical data. The main objective of the study is to evaluate the effectiveness of the EMAeHealth digital tool, and assess its usability and acceptability under routine conditions.

**Methods and analysis** Hybrid implementation-effectiveness design: (1) A cluster randomised, prospective, longitudinal, multicentre clinical trial to evaluate the effectiveness of EMAeHealth in (A) improving health-related quality of life (primary outcome), (B) improving self-efficacy for labour and childbirth and self-efficacy in breast feeding and (C) reducing the number of visits to the obstetric emergency services and health centre in situations of 'non-pathological pregnancy', 'false labour pains' and 'non-pathological puerperium'. The EMAeHealth intervention plus usual care will be compared with receiving only usual care, which includes traditional ME. N=1080 participants, 540 for each study arm. Two measurements will be made throughout the pregnancy and three in the first 16 weeks post partum. (2) A mixed-method study to evaluate the usability and acceptability of the tool, barriers and facilitators for its use, and implementation in our health system: focus groups (women, professionals and agents involved) and a quantitative analysis of implementation indicators. Analysis: It will be carried out by intention to treat, using mixed models taking into account the hierarchical structure of the data and per protocol to evaluate the effectiveness of the express use of the digital tool.

**Ethics and dissemination** Clinical Research Ethics Committee of Euskadi, Spain, (Ref: PI2020044) approved this study. The results will be actively disseminated through manuscript publications and conference presentations.

**Trial registration number** NCT04937049.

## STRENGTHS AND LIMITATIONS OF THIS STUDY

⇒ EMAeHealth is the prototype of a digital tool that supports decision-making during the pregnancy and post partum. One strength is that it has been designed through Collaborative Action Research and is intended to be a complement to Maternal Education.

⇒ A hybrid efficacy-implementation design will allow the effectiveness of EMAeHealth to be evaluated while collecting data on the implementation of the tool in 'real-world' conditions.

⇒ Both patients and professionals have participated in the previous Collaborative Action Research and will be involved in the evaluation of the tool and in its implementation.

⇒ One of the main strengths of the efficacy-implementation study is that it will result in an effective tool that will be perfectly adapted to the needs of the target population.

⇒ A weakness is that it is possible the results cannot be directly extrapolated to a different context and it would be necessary to adapt the tool to the context where it will be implemented.

## INTRODUCTION

### Need to adapt maternal education to the current context

Antenatal education classes, including 'childbirth education programmes', 'prenatal classes' and antenatal groups, are attended by a large percentage of pregnant women worldwide.[1–3] Both the high attendance of maternal education (ME) classes, and the fact that they take place at a time of transition, when readiness for change is higher, suggest that ME can be a valuable window of opportunity.[1 2] However, the benefits of these programmes for participants and their newborn infants remain unclear, and classes are not normally based on the expressed needs of attendees, but rather on the messages that the educators

themselves believe they should impart.[1] It may be that studies aimed at evaluating the effectiveness of ME have tended to have methodological weaknesses, such as small sample sizes, lack of control for variables that could affect the delivery process and consideration of only part of the delivery process.[1] Whatever the case, it has also been found in our setting that ME classes have not been associated with an improvement in most birth variables, whether physical or psychological, and have even increased birth anxiety in foreign women[3]; in this study, it was concluded that a new approach to these classes would be necessary. Other authors have come to the same conclusion, in light of their results.[4 5] There is therefore a need to reorient ME by focusing on women's needs and to evaluate the effectiveness of any initiatives to improve ME.

In order to establish a feasible and effective perinatal education programme, tailored to our setting, the team—following guidance published by the UK Medical Research Council for the development and evaluation of complex interventions in primary care—undertook: (A) a qualitative study with 30 women, exploring their needs during pregnancy and post partum; (B) two literature reviews on women's needs at these times and theoretical models of health education and (3) seven discussion and consensus sessions with an expert panel of midwives, gynaecologists, paediatricians, paediatric and postpartum nurses.[6 7] The aim was to involve women and professionals in the process of designing the new ME. Subsequently, a study was conducted to establish priorities with respect to ME using the Delphi technique and the nominal group approach. The objective of this study was to identify and prioritise the most important issues in ME in order to set goals and evaluate their achievement, that is, to 'make ME evaluable'.[8]

The needs expressed by women in our population changed throughout pregnancy: initially they needed information to know that 'everything was going well'; next, they needed more emotional support, to feel confident and self-reliant in dealing with their fears about childbirth and childcare; and finally, they needed more postpartum support and less pressure regarding breast feeding. They also preferred a comprehensive perinatal education programme—covering pregnancy, post partum, breast feeding and parenting—rather than just prenatal; and one that was more participatory and flexible and allowed for greater partner involvement. These were the results of the study with focus groups of women.[6]

Accordingly, ME was redefined as a complex intervention, which should extend from the beginning of pregnancy to the end of the puerperium, focusing on the needs of women and with the involvement of multiple health professionals. It would be an intervention whose general objective should be to enable women to make appropriate decisions to: (1) improve their health level and that of their family, (2) choose and responsibly face their birth and parenting model, and (3) maintain family and social support networks. The new model of ME includes multiple interventions capable of responding to the health needs of each woman at different moments of the childbearing process.[7 8]

## Design and relevance of an eHealth tool to support ME: EMAeHealth

EHealth interventions are becoming increasingly important and frequent.[9] EHealth can help the health service user to take a more active role in decision-making and enable the health professional to guide the patient in this process.[10] It can also contribute, at organisational level, to a more patient-centred model of healthcare.[11]

In addition, young women from all social strata consistently use the Internet as a source of information about their health.[12–15] The information available on the websites they consult is positively correlated with the decisions they make about their health during the pregnancy and post partum.[12 16] Pregnant and postpartum women use the internet because it offers immediate, detailed, entertaining, personalised or practical information.[13 17]

However, the lack of participation of health professionals in integrating healthcare and new technologies is felt,[12 18] since the currently available range of information is not always reliable[19] or contextualised.[20] As the origin of a website has a direct effect on reliability, the participation of health professionals in the use, advice and generation of new technologies is essential.[15 21]

This team has designed a prototype digital tool called EMAeHealth involving healthcare professionals and patients, using a collaborative research process.[21] The website, adapted to our context of implementation, is organised into four areas: (1) an information area, which is open to the public, provides information based on evidence and is permanently updated by healthcare professionals; (2) a communication area, which allows women to contact others in a similar situation, through forums or conversations, or consult their own midwife (3) a health self-management area, which has valid and reliable self-assessment instruments for checking or reflecting on their own health needs, and decision support tools such as the detection of the onset of labour or the detection of warning signs and symptoms in the post partum; and (4) a clinical data area, which allows women to keep their clinical data, to add the most recent, and share data with other professionals if they wish, under secure conditions. The tool is conceived as a complement that can reinforce ME; with resources that facilitate its accessibility, immediacy and continuity, as well as improving its results. Its incorporation into the Osakidetza-Basque Health Service corporate website will encourage its adoption by patients and professionals, as well as its safe use. The usual phases for the development of a digital tool are analysis, development, testing and production. The prototype is the result of the analysis, a proposal of the architecture and technical requirements that the digital tool must have in order to respond to the users' needs. This is followed by development—or programming—and testing, which are carried out almost simultaneously, to detect and solve errors. Once the tool is ready, it is deployed in a

production environment, making it available to all users of the system, in our case the Osakidetza-Basque Health Service corporate website.

## Study of the effectiveness of the tool while we are collecting useful information for its implementation in routine conditions

The evaluation of the effectiveness of digital tools aimed at health education during pregnancy is scarce and has had controversial results:[22 23] the quality, reliability and effectiveness of current pregnancy apps is yet to be determined,[15 22 23] and consequently, such studies to evaluate effectiveness are absolutely essential, especially if the digital tool comes from the healthcare system. In addition, there are barriers and facilitators that modify both the use of digital technology and its effectiveness in healthcare, where further study is needed.[13 16 17 24–26] Fortunately, there are hybrid designs that enable testing of the clinical intervention and, at the same time, during the trial, gathering information on its application and/ or its potential for application in real-world conditions.[27] There are also frameworks, such as the RE-AIM (Reach, Effectiveness, Adoption, Implementation, Maintenance) Planning and Evaluation Framework, and Consolidated Framework for Implementation Research (CFIR), which can help in the task of studying the degree of adoption of an innovation, and guide in the data collection, analysis and interpretation of barriers and facilitators to implementation.[28 29] Therefore, we conducted this study to find out if complementing traditional perinatal care with a tool—designed through a collaborative action-research process—translates into better health outcomes for women than those obtained with traditional care alone. In addition, we want to know if this type of tool is useful and acceptable to its users in real-life conditions.

## OBJECTIVES

The main objective of the study is twofold: on the one hand to evaluate the effectiveness of the EMAeHealth digital tool and its components (Information Area, Communication Area, Health Self-Management Area and Clinical Data Area), and on the other to assess the usability and acceptability of the tool under routine conditions.

The following are proposed as specific objectives:

► To develop, test and deploy the EMAeHealth tool,[21] with the structure described above, capable of offering information, responses via algorithm, links to available resources, secure data archiving and monitoring, connection between various systems and the possibility of data analysis establishing continuous feedback and readjustment of the information presented. To provide it with content related to pregnancy, childbirth and post partum.

► To evaluate the effectiveness of the tool in (1) improving health-related quality of life, (2) improving self-efficacy for labour and childbirth and self-efficacy in breast feeding and (3) reducing the number of visits to the obstetric emergency services and health centre

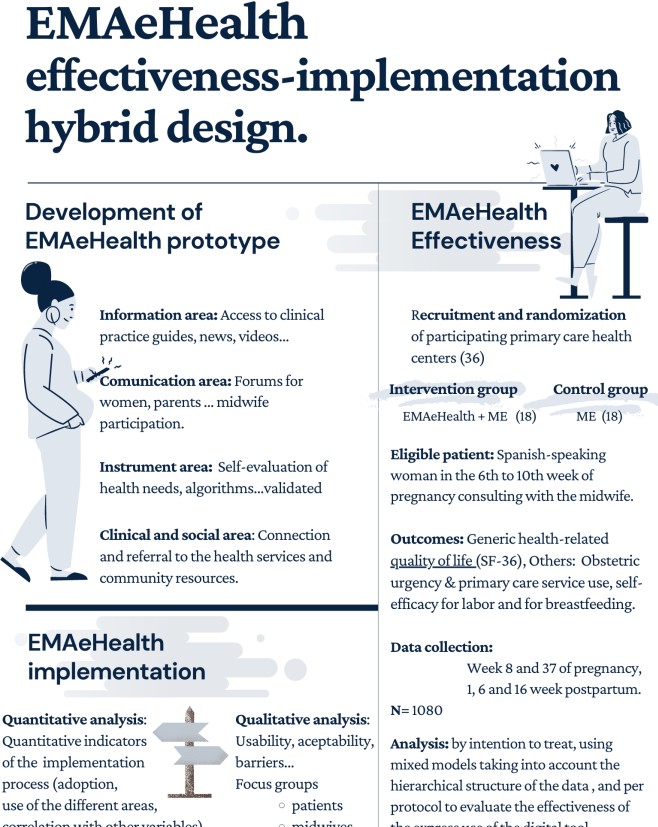

**Figure 1** EMAeHealth design description. ME, maternal education; SF-36, Short Form 36.

in situations of 'non-pathological pregnancy', 'false labour pains' and 'non-pathological puerperium'.

► To evaluate the usability and acceptability of the tool by women and professionals, and any barriers to its implementation.

► To estimate the differential impact that EMAeHealth could have, based on sociodemographic and/or population characteristics.

## METHODS AND ANALYSIS

This is an Effectiveness-implementation hybrid design type I,[26] which combines (1) a multicentre, longitudinal, prospective, cluster randomised clinical trial to evaluate the effectiveness of EMAeHealth and (2) a mixed-method study to evaluate the usability and acceptability of the tool, barriers and facilitators for its use and implementation in our health system. The mixed-method study is made up of a qualitative study by focus groups (women, professionals and agents involved) and a quantitative analysis of implementation indicators (figure 1).

For the effectiveness study, we have chosen a cluster randomised design to avoid, among other issues, contamination bias among women from the same centre, since they are likely to coincide, for example, in ME classes. In addition, as the intervention has been designed for shared decision-making, it did not seem appropriate

for the same midwife to have women belonging to both control and intervention.

The study is being conducted between 2021 and 2023, with the first year dedicated to the development of the EMAeHealth digital tool. The general structure of the tool is described in the introduction. In the development of the prototype, the International Patient Decision Aid Standards were applied, using a collaborative action research process, involving experts and patients, with qualitative research techniques, as well as focus groups, prioritisation and consensus techniques.[21] At this time, we are finalising content editing and software development and testing. We expect to start recruitment in July 2022. The participants will be followed up for a year, the period of time that the intervention lasts. The final months of 2023—first months of 2024 will be dedicated to the analysis and diffusion of results.

Regarding the study population, all Osakidetza-Basque Health Service public primary care centres that have midwives who are interested in optimising their clinical practice by adopting innovations will be eligible. In order to consider the centre eligible, the requirement will be that at least 50% of the midwives belonging to the centre have signed a collaboration agreement, as well as the approval of the Head of the Primary Care Centre.

Osakidetza-Basque Health System is the public healthcare system of the Basque Country, a region located in the north of Spain. Osakidetza was created by the Health Department of the Basque Government in 1983. All the public hospitals and primary care centres of the Basque Region are under this organisation, structured into several Integrated Health Organisations (IHO) spread throughout the Basque country. More than 30 000 professionals work for Osakidetza, and it could be considered the largest organisation in the Basque Country.

For the study of clinical effectiveness, an active surveillance system will be established to identify eligible patients, who are Spanish-speaking pregnant women attending their first appointment with the midwife between week 6 and 10 of gestation. Regarding their clinical profile, they are low-risk pregnant women. The midwives participating in the study will be in charge of recruiting them consecutively until they have the necessary sample from among the people using and not using the EMAeHealth tool (figure 2).

Regarding the mixed-method study, to evaluate the quantitative implementation indicators, all pregnant women who have seen their midwives at the start of gestation at the health centres will be included for 1 year after the implementation of EMAeHealth. The same goes for the qualitative evaluation, through focus groups of women and professionals from the health centres (model consent form as an online supplemental file 1).

### Random assignment of centres
The eligible centres will be randomised centrally, blindly, and electronically, in the Biocruces-Bizkaia Research Institute, stratified by IHO and in a 1:1 ratio. Therefore,

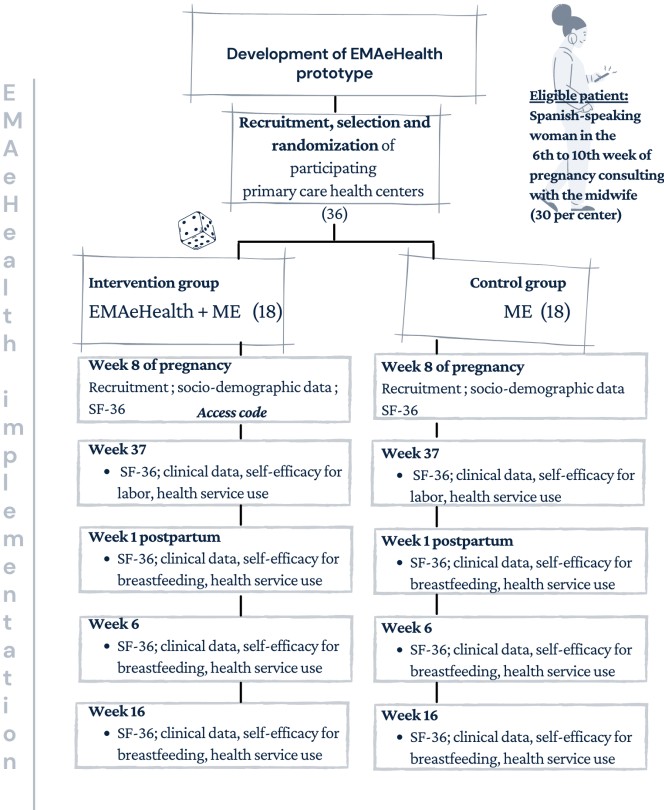

**Figure 2** EMAeHealth effectiveness flow chart. ME, maternal education; SF-36, Short Form 36.

the intervention allocated centres will have access to EMAeHealth, while it will not be accessible for the control group centres. IHOs are groups of health centres within the same geographical area, with similar characteristics and population, under the direction of the same hospital. Therefore, we assume that there are no pre-existing differences in the quality of care provided between centres that belong to the same IHO.

### Intervention centres
All the patients in these centres, in the first consultation with the midwife, will be informed that EMAeHealth is available to them, and they will be told about the content of the tool. In addition, they will be invited to participate in the study of its clinical effectiveness. If the answer is affirmative, they must sign the consent form and will be provided with a link to the forms (baseline measurements and questionnaires for each measurement) to fill in electronically. Participants sign a consent form approved by the Ethics Committee and can leave the study whenever they want (model consent form as an online supplemental file 1).

### Control centres
Patients in these centres will not have EMAehealth. They will receive, like the intervention group, the usual care and the possibility of attending ME sessions, but not

EMAeHealth. After being informed of the clinical effectiveness study, they will be asked for consent to participate (see online supplemental file 1). Women who sign the consent form for participation in the study will be provided with the link to the forms for each measurement.

## Outcomes

Access to EMAeHealth by women and professionals during the study period will be through access codes, which will allow the research team to track the use made by users of the tool while ensuring the confidentiality of the data. It also makes it possible, under safe conditions, to link this use with clinical variables from medical records and with the record of the forms filled in electronically at each measurement.

Results of clinical effectiveness: The primary outcome is change in Health-related quality of life (QoL), measured with the Spanish version of the Short Form 36 (SF-36) questionnaire.[30] The secondary variables are the number and reasons for visits to the obstetric emergency service and to the health centre for non-pathological pregnancy situations, and the labour stage at the time of going to the hospital for suspected labour onset (measurement is carried out using the Bishop test/dilation in cm, effacement and breaking of waters (YES/NO)) . As indirect indicators of autonomy in managing their own health and well-being, self-efficacy for labour and childbirth and self-efficacy in breast feeding will be measured using the Spanish version of Lowe's CBSEI scales[31] and the Breastfeeding Self-Efficacy Scale—SF,[32] respectively.

Measurements of these variables will be performed at weeks 8 and 37 of pregnancy, and at weeks 1, 6 and 16 post partum. Contact with the woman and the response to the questionnaires will be carried out electronically (via mobile phone).

QoL is a parameter that has been defined and recently taken into account as a health indicator by WHO.[33] The concept of QoL could be defined as an individual's assessment of their own state in relation to their needs,[33] so it seems appropriate that it should be the primary outcome measure. The Spanish version of the SF-36 questionnaire has been used to assess the health-related QoL perceived by pregnant women in several studies,[33 34] showing good psychometric characteristics. The SF-36 questionnaire is made up of eight domains, of which four are oriented towards physical aspects and another four towards mental aspects of health-related QoL. In addition, two summaries are made of this physical and mental component, which explain between 80% and 85% of the variance of the eight domains.[30] The scores of the questionnaire are transformed on a scale from 0 to 100 points; the higher the score, the better the perception of health-related QoL. A difference of five points in the domain score, or between 2 and 3 points in the summary score is considered clinically relevant.[30]

Self-efficacy is one of the key factors influencing women's confidence and ability to cope with childbirth, and the Childbirth Self-Efficacy Inventory has been shown to be a valid and reliable instrument to measure maternal confidence in childbirth. The Spanish version of the Childbirth Self-Efficacy Inventory shows adequate psychometric properties (ie, internal consistency and validity). Moreover, self-efficacy is a variable which is sensitive to external modifications, so we believe it may be suitable as one of the secondary measures to assess the effectiveness of using EMAeHealth.

Similarly, breastfeeding self-efficacy is a mother's confidence in her ability to breastfeed and is highly predictive of breastfeeding behaviours. The Spanish version of the BSES-SF can be considered a valid and reliable measure of maternal breastfeeding self-efficacy.

Indicators of the implementation process: Qualitative indicators: Usability and acceptability of the tool by patients and professionals, as well as barriers and facilitators for its use and adoption in the health system. The qualitative evaluation will be carried out through focus groups of women and professionals, between 6 and 8 per group and as many as necessary until the discourse is saturated.[35] A semistructured script will be developed for the sessions, based on the theoretical constructs of the CFIR,[28] which will also serve as the basis for the analysis. The CFIR includes 37 constructs within 5 main domains: intervention characteristics, outer setting, inner setting, individual characteristics and implementation process. Overall, the CFIR aims to help identify potential factors (ie, barriers and facilitators) that are believed to influence implementation. Sessions will be audiorecorded and moderated by an expert in qualitative techniques while an assistant will take notes. Later the sessions will be transcribed and analysed. Quantitative indicators: adoption and frequency of the use of EMAeHealth and its areas according to stratification profiles and correlates with other variables (indicators: patients exposed to the digital tool, by centre and group; women with access to EMAeHealth who access the tool once; frequency of use of the tool, frequency of use of the different functions of the tool, etc).

Predictive, modifying and confounding variables: Patients: Age, parity, nationality, race, social class, educational level, marital status, work situation, distance to hospital, spontaneous pregnancy versus assisted reproductive technique, obstetric history and other clinical characteristics. Professionals: Sex, age, length of time in the position, type of contract, profile of adoption of innovations by professionals, measured by means of a questionnaire by Borracci et al, 2013.[36] Health centres: reference IHOs, number of midwives and participation rate, rural or urban environment, and population served (figure 3).

## Statistical methods

EMAeHealth clinical effectiveness results: quantitative comparisons will be made between intervention and control centres, with an intention-to-treat approach. Mixed models (linear or generalised as appropriate to

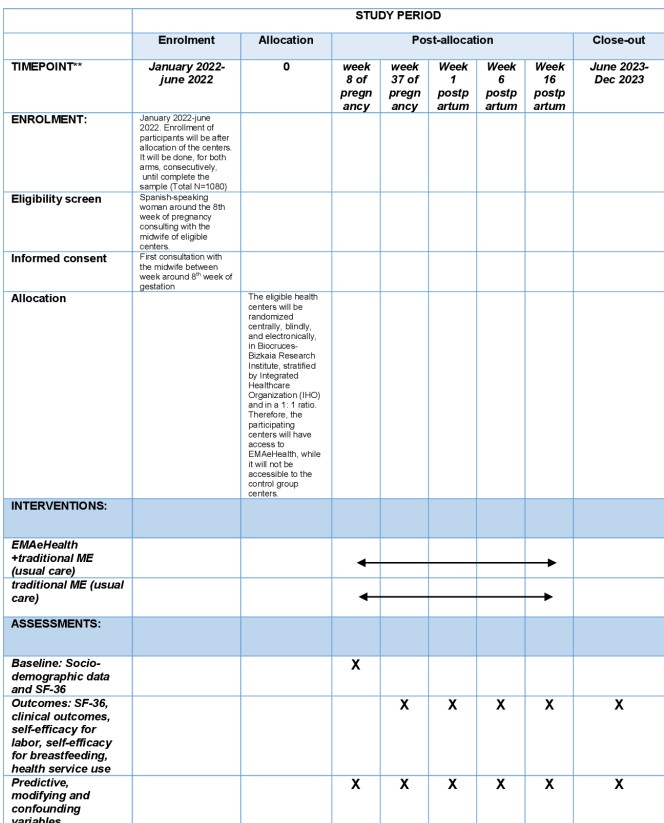

| | STUDY PERIOD | | | | | | | |
|---|---|---|---|---|---|---|---|---|
| | Enrolment | Allocation | Post-allocation | | | | | Close-out |
| **TIMEPOINT**** | *January 2022-june 2022* | 0 | week 8 of pregnancy | week 37 of pregnancy | Week 1 postpartum | Week 6 postpartum | Week 16 postpartum | *June 2023-Dec 2023* |
| **ENROLMENT:** | January 2022-june 2022. Enrollment of participants will be after allocation of the centers. It will be done, for both arms, consecutively, until complete the sample (Total N=1080) | | | | | | | |
| **Eligibility screen** | Spanish-speaking woman around the 8th week of pregnancy consulting with the midwife of eligible centers. | | | | | | | |
| **Informed consent** | First consultation with the midwife between week around 8th week of gestation | | | | | | | |
| **Allocation** | | The eligible health centers will be randomized centrally, blindly, and electronically, in Biocruces-Bizkaia Research Institute, stratified by Integrated Healthcare Organization (IHO) and in a 1:1 ratio. Therefore, the participating centers will have access to EMAeHealth, while it will not be accessible to the control group centers. | | | | | | |
| **INTERVENTIONS:** | | | | | | | | |
| *EMAeHealth +traditional ME (usual care)* | | | ←————————————→ | | | | | |
| *traditional ME (usual care)* | | | ←————————————→ | | | | | |
| **ASSESSMENTS:** | | | | | | | | |
| *Baseline: Socio-demographic data and SF-36* | | | X | | | | | |
| *Outcomes: SF-36, clinical outcomes, self-efficacy for labor, self-efficacy for breastfeeding, health service use* | | | | X | X | X | X | X |
| *Predictive, modifying and confounding variables* | | | X | X | X | X | X | X |

**Figure 3** SPIRIT timetable. ME, maternal education; MS, maternal education; SF 36, Short Form 36; SPIRIT, Standard Protocol Items: Recommendations for Interventional Trials. **This period is approximate. It depends on the availability of EMAeHealth on the Osakidetza-Basque Health Service corporate website.

the variable under study), which take into account the hierarchical structure of the data (patients grouped by midwives, midwives grouped by centre), will be used and 95% CIs will be calculated. Likewise, per-protocol analysis will be carried out, in which the effectiveness of the express use of both the tool in general and in particular of some of its components will be evaluated, adjusting for possible confounding factors. Statistical analysis will be carried out by a professional who is not part of the research team.

Implementation process: For the qualitative evaluation of the tool and its implementation, the analysis of the focus groups, facilitated by the ATLAS.ti software programme, will be structured in three phases: a deductive analysis to determine which CFIR constructs influence its use in real life, followed by an inductive analysis to identify others not included in the CFIR framework that could influence its use, and a third validation phase using triangulation. The analysis results report will be provided to each patient and professional, who will have the opportunity to provide feedback to the team of analysts with their comments ('response validity').

For the quantitative evaluation, the distribution of the indicators associated with the use and scope of the tool as well as its components will be described. Measures of central tendency, dispersion and proportions will be calculated, with 95% CIs, and the relationship between the indicators and the predictor/confounding variables will be analysed using mixed models.

## Study size

For the clinical effectiveness study, 18 centres will need to recruit 30 patients each, in each arm of the study (N=1080). This will make it possible to detect a difference of two points as significant in the mental component of the SF-36 test, with an alpha error of less than 5% and power greater than 80%. A 10-point SD is assumed for the mental SF-36 and an intracluster correlation of 0.01.

The calculation took into account the cluster design of the study and it was carried out using the National Institute of Health GRT sample size calculator* which assumes that the analysis will employ mixed-model regression methods.

* Research Methods Resources: National Institutes of Health. (Accessed on 2 January 2020). Available from: https://researchmethodsresources.nih.gov/grt-calculator

## Patient involvement

In the previous studies, multidisciplinary groups of midwives, nurses, paediatricians, epidemiologists, psychologists and pregnant and postpartum women were involved in the description and prioritisation of ME needs, as well as in the redesign of the ME and the design of an EMAeHealth tool to complement it.[3 6–8 8 15 21] In this study, patients will be involved in testing the tool under construction, assessing its usability, acceptability and evaluating barriers and facilitators for its implementation in 'real-world conditions'. In addition, women will participate as study subjects in the evaluation of the effectiveness of the digital tool.

## ETHICS AND DISSEMINATION

The study protocol has been reviewed and ethics approval obtained from the by the Clinical Research Ethics Committee of Euskadi, Spain in November 2020 (Ref: PI2020044). This approval covers all centres included in the study. Participation in this study will be voluntary; individual written (or online) consent will be sought from all participants. Each participant will receive a study information sheet that outlines the project and expectations in plain language

**Author affiliations**
[1]Primary Care Subdirectorate, Osakidetza-Basque Health Service, Vitoria-Gazteiz, Spain
[2]Primary Health Care Research Group, Biocruces Bizkaia Health Research Institute, Barakaldo, Spain
[3]Zuazo Primary Care Health Centre, Osakidetza-Basque Health Service, Bizkaia, Spain
[4]School of Nursing, University of the Basque Country, Barakaldo, Spain
[5]Sestao Primary Care Health Centre, Osakidetza-Basque Health Service, Bizkaia, Spain

⁶Midwifery Training Unit, University of Basque Country, Barakaldo, Spain
⁷PB, Methodological and Statistical Consulting, Bizkaia, Spain
⁸Biocruces Bizkaia Health Research Institute, Barakaldo, Spain

**Acknowledgements**  We would like to acknowledge the collaboration of those interested in the design of the EMAeHealth tool, the grant received by the Carlos III Health Institute in collaboration with the ISCII-Subdirectorate General for Evaluation and Promotion of Research (European Regional Development Fund-FEDER) and the support of Biocruces Bizkaia Health Research Institute. We also want to thank the invaluable work of Katy Ryan, who has been responsible for translating the manuscript into English.

**Collaborators**  ema-Q Goup is a multidisciplinary group made up mainly of Basque Health service midwives, as well as other medical professionals, psychologists, quality technicians and research experts: Itziar Estalella (IE), Mª José Trincado (MJT), Inés Cabeza (IC), Mari Pierre Gagnon (MPG), Ana Fernández (AF), Gorane Lozano (GL), Gemma Villanueva (GV), Jesús Sánchez (JS), Amaia Maquibar (AM), David Moreno (DM), Catalina Legarra (CL), Maria Jesús Mulas (MJM), Mónica Blas (MB), Pilar Amorrortu (PA), Sonia Alvarez (SA) and Covadonga Pérez (CP).

**Contributors**  IA-P, CP-P, PB-G and MEC designed the EMAeHealth program, with EMA-Q Group's collaboration, AG-A has collaborated on the statistics and on the revision of the article; All of them have been involved in the design of the protocol and have collaborated in writing the manuscript; all authors read, contributed and finally approved the manuscript. Contributors: AG-A (statistics) and patients involved.

**Funding**  This study has been funded by the Carlos III National Health Institute, (grant number: PI20/00899.), (Co-funded by European Regional Development Fund 'A way to make Europe').

**Competing interests**  None declared.

**Patient and public involvement**  Patients and/or the public were involved in the design, or conduct, or reporting, or dissemination plans of this research. Refer to the Methods section for further details.

**Patient consent for publication**  Not applicable.

**Provenance and peer review**  Not commissioned; externally peer reviewed.

**ORCID iD**
Maite Espinosa Cifuentes http://orcid.org/0000-0003-4886-3270

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
