## [Reviewer comments · BMJ Open]

ARTICLE DETAILS

TITLE (PROVISIONAL)	EMAEHealth, a digital tool for the self-management of women's health needs during pregnancy, childbirth and the puerperium: protocol for a hybrid effectiveness-implementation study.
AUTHORS	Espinosa Cifuentes, Maite; Artieta-Pinedo, Isabel; Paz-pascual, Carmen; Bully-Garay, Paola; García-Alvarez, Arturo; Group, emaQ

VERSION 1 – REVIEW

REVIEWER	Helen Cheyne University of Stirling, NMAHP Research Unit
REVIEW RETURNED	19-Sep-2021

GENERAL COMMENTS	Thank you for inviting me to review this study protocol. The topic of digital information, health promotion, decision-making and communication for women through pregnancy and following birth is important and relevant to readers of this journal. However I feel that some revision is required before it is suitable for publication. Overall, it is difficult to read and to follow. I think that this is because English is not, or does not appear to be, the first language of the authors. Editing the manuscript to correct language may help the understandability of the paper. However, I feel that some reorganisation and clarification is required in some sections. The submission includes a completed SPIRIT checklist however, I feel that adhering to the SPIRIT guidance in ordering and in enhanced descriptions would improve to paper considerably Trial design (spirit-statement.org). Specific comments: Introduction The rationale in the introduction needs further explanation and justification through inclusion of a summary of relevant studies and clearer explanation of the background and context of the study. I am not sure that it is a fact that nearly all pregnant nulliparae attend maternal education (possibly this is the case in a few countries). Paragraph 2 states - Taking into account the bibliography and the needs expressed in focus groups by pregnant and postpartum women in our environment,- this requires more explanation for example what bibliography is referred to? Was a literature review undertaken by this team? More needs to be said about the focus groups for example was their purpose to develop the intervention? Line 47 – I don't think that 'train' women is the most appropriate word to use here, possibly educate or enable would be better. Page 4 line 39 refers to context – what is the context? Line 45 or consult their reference professional – does this mean their own doctor or midwife? Line 51 -as the detection of the moment of delivery – does this mean the onset of labour? Reading the introduction it seems that a website for maternal
---

	education has been developed based on a literature review and consultation with women via focus groups. The website has been adopted in clinical practice but requires to be evaluated while in situ. However, later in the methods sections it seems that the intervention will be developed in the first stage of the study. This is confusing and needs to be clarified. Page 7 line 13 The evaluation of the effectiveness of digital tools in education for health in pregnancy is scarce and has had controversial results [22,23], which makes it essential. This sentence needs reworded for clarity. Aim – is clear but implies that the tool is already developed – given the first objective is to develop the tool this should be reflected in the AIM. Objective 1 is also inconsistent with the introduction which suggests the tool is already developed and in use. The development of a tool such as this is in itself a substantial piece of work that requires more description. The SPIRIT guidance makes clear that if important elements of interventions are not described (in particular complex interventions) it is impossible for reviewers to fully understand the trial. The SPIRIT checklist suggests that the Design section should follow the objectives and I feel that conforming to this would improve the clarity of the paper considerably. Page 8 line 29 provides a statement of the design, however there should be more of an explanation of the key points in a multicenter, longitudinal, prospective, cluster-randomized clinical trial, for example why a cluster design was chosen and what are the main elements. Further, there is a design phase (objective 1) this requires description. The study setting eligibility criteria and participants are described in rather vague terms. This is particularly important in a cluster trial where clarity about issues such as the unit of randomisation and levels of participants for example- maternity clinic, midwife, pregnant woman, is essential. I presume that the primary care centres are the clusters, is consent to participation of the cluster obtained from the head of the Primary Care Centres. Outcomes The chosen outcomes seem relevant and interesting but they require more detailed description and justification for their use in this study. Indicators of the implementation process. This is important but under described. Why have focus groups been chosen rather than individual interviews? Will focus groups be undertaken in control sites – it might be important to know more about women’s information seeking and access to maternal education, including commercially available Apps. A semi-structured script will be developed based on the theoretical constructs of the CFIR - what are these? The participant flowchart is helpful. Study size, I am not able to assess the information provided here. I would be interested to see some more information that acknowledges that this is a cluster trial and the implications of that both for the calculation of the sample size and for the subsequent analysis of data. Page 13 Patient involvement. This section describes a previous study where a multidisciplinary group prioritised needs for Maternal Education. I think this may be what was alluded to in the introduction. If so, it would be helpful to have this work more fully described in the introduction. Strengths and limitations The first three are statements which one may agree with but are not direct strengths and limitations of this study
--	--

REVIEWER	Neeltje Crombag
REVIEW RETURNED	21-Jan-2022

GENERAL COMMENTS	Thank you for giving me the opportunity to review your interesting study protocol. The topic and application is relevant and I would look forward to see the outcomes of this project. Below some suggestions to maybe improve your study protocol Strength and limitations Here I would suggest rephrasing the sentences in which it becomes truly clear where the strengths and limitations are and why. For example: EHealth can help the user of health services to take a more active role in decision-making and serve the professional to guide the patient in this process. It is a (major) strength of this study to include patients in the design and evaluation of these new technologies as this.....etc Introduction: First sentence should be split in 2 or 3 sentences Methods and analysis: It would be helpful if you could be a bit more detailed on the design of the tool. How is it designed, by which method (which design method is used) and by whom (are patients and healthcare professionals involved)? Also can you specify the inclusion criteria related to their clinical profile for eligible patients? Are they all low-risk at the beginning of pregnancy (assuming this as under midwifery care) but might be good to specify this. If I understand correctly, the tool is located randomly to a centre: so if centre A is allocated, centre A will use the ehealth, while centre B would not. But how can you be sure there are no pre-existing differences in the (quality of) care provided between centres? Or can you indicate why they are comparable? Also do all these centres serve the same population? Sometimes there are differences between because of location or the population they serve.
--

VERSION 1 – AUTHOR RESPONSE

Reviewer: 1

Dr. Helen Cheyne, University of Stirling Comments to the Author:

Thank you for inviting me to review this study protocol. The topic of digital information, health promotion, decision-making and communication for women through pregnancy and following birth is important and relevant to readers of this journal. However I feel that some revision is required before it is suitable for publication. Overall, it is difficult to read and to follow. I think that this is because English is not, or does not appear to be, the first language of the authors. Editing the manuscript to correct language may help the understandability of the paper. However, I feel that some reorganisation and clarification is required in some sections. The submission includes a completed SPIRIT checklist however, I feel that adhering to the SPIRIT guidance in ordering and in enhanced descriptions would improve to paper considerably Trial design (spirit-statement.org).

- First of all, thank you very much for your suggestions. Regarding the quality of the English, we have revised the protocol again with the help of a native English colleague; we hope that, after the changes made, the protocol will be more understandable.

Regarding adherence to the SPIRIT guidelines, we believe it is difficult to adapt the protocol strictly to the guideline due to the nature of the study design. We have attempted to describe, in parallel in each

of the protocol sections, on the one hand the clinical trial used to measure the efficacy of EMAeHealth, and on the other hand the mixed methods study to measure the usability and acceptability of the tool. We believe that the clarifications we have added will make the manuscript more understandable.

Specific comments:

Introduction

The rationale in the introduction needs further explanation and justification through inclusion of a summary of relevant studies and clearer explanation of the background and context of the study.

- We sincerely believe that you are absolutely right in this. In writing the protocol, we made an error in a bibliographic citation (21), which we believe has conditioned both the understanding of the context of the study and the stage we are at. This article 21 describes the steps taken to design the EMAeHealth tool. It reports that, prior to the study referred to in this protocol, extensive research was carried out on the needs of women in terms of Maternal Education, from the point of view of the women themselves and professionals, and an analysis of the evidence on needs, models of MS, sources of information most used on the Internet by women, etc. We have modified this bibliographical citation and clarified the first part of the introduction.

(21) Artieta-Pinedo I, Paz-Pascual C, Bully P, Espinosa M, EmaQ Group. Design of the Maternal Website EMAeHealth That Supports Decision-Making During Pregnancy and in the Postpartum Period: Collaborative Action Research Study. *JMIR Form Res* 2021;5(8):e28855. URL: <https://formative.jmir.org/2021/8/e28855>
DOI: 10.2196/28855

I am not sure that it is a fact that nearly all pregnant nulliparae attend maternal education (possibly this is the case in a few countries).

- In our context, we have seen that 92% of nulliparas attended classes, according to a study we carried out in 2008 with a sample of 600 participants (*). However, you are right when you state that this is only the case for a few countries, so we have omitted this statement.

(*)Paz-Pascual C, Pinedo IA, Grandes G, de Gamboa GR, Hermosilla IO, de la Hera AB, Gordon JP, Garcia GM, de Pedro MU. Design and process of the EMA Cohort Study: the value of antenatal education in childbirth and breastfeeding. *BMC Nurs*. 2008 Apr 24;7:5. doi: 10.1186/1472-6955-7-5. PMID: 18435856; PMCID: PMC2386782.

Paragraph 2 states - Taking into account the bibliography and the needs expressed in focus groups by pregnant and postpartum women in our environment,- this requires more explanation for example what bibliography is referred to? Was a literature review undertaken by this team? More needs to be said about the focus groups for example was their purpose to develop the intervention?

- Done.

Line 47 – I don't think that 'train' women is the most appropriate word to use here, possibly educate or enable would be better.

- Agreed

Page 4 line 39 refers to context – what is the context?

- By context, we mean the set of circumstances or unique factors in which implementation takes place, for example, an organisation, a community, or the wider system. (Pfadenhauer et al., 2017). We have replaced "context" with "Context of implementation"

Line 45 or consult their reference professional – does this mean their own doctor or midwife?

- Midwife. We have now indicated.

Line 51 -as the detection of the moment of delivery – does this mean the onset of labour?

- Yes. We have replaced "as the detection of the moment of delivery " with "onset of labour"

Reading the introduction it seems that a website for maternal education has been developed based on a literature review and consultation with women via focus groups. The website has been adopted in clinical practice but requires to be evaluated while in situ. However, later in the methods sections it seems that the intervention will be developed in the first stage of the study. This is confusing and needs to be clarified.

- We have added a paragraph to clarify this issue :

“The usual phases for the development of a digital tool are analysis, development, testing and production. The prototype is the result of the analysis, a proposal of the architecture and technical requirements that the digital tool must have to respond to the needs of the users. After this stage begins the development, or programming, and the testing, which are done almost simultaneously to detect and solve errors. Once the tool is ready, it is deployed in a production environment, making it available to all users of the system, in our case the Osakidetza-Basque Health Service corporate website.”

Page 7 line 13 The evaluation of the effectiveness of digital tools in education for health in pregnancy is scarce and has had controversial results [22,23], which makes it essential. This sentence needs reworded for clarity.

- Sorry for our English. We really try to do the best we can... We have replaced this sentence with: “The evaluation of the effectiveness of digital tools aimed at health education during pregnancy is scarce and has had controversial results [22,23]; consequently, its realization becomes absolutely essential”

Aim – is clear but implies that the tool is already developed – given the first objective is to develop the tool this should be reflected in the AIM. Objective 1 is also inconsistent with the introduction which suggests the tool is already developed and in use.

- The main objective of the study is to evaluate the effectiveness of the EMAeHealth digital tool, as well as its usability and acceptability under routine conditions. The first specific objective is the development of the tool. When we say "development" we are not talking about the design of the intervention (that would be the analysis phase, which is already done), but about the "manufacturing", or "building" the prototype already designed(21), and uploading EMAeHealth to the web platform of the Osakidetza health service. To make the objective clearer, we have replaced "To develop EMAeHealth..." with "Develop, test and deploy the EMAeHealth tool...", according to the phases described in the paragraph just included.

The development of a tool such as this is in itself a substantial piece of work that requires more description. The SPIRIT guidance makes clear that if important elements of interventions are not described (in particular complex interventions) it is impossible for reviewers to fully understand the trial.

- We believe that now that the mistake in the bibliographic citation, which refers to the design of the intervention, has been corrected, the reader will have no problem understanding the intervention. The SPIRIT checklist suggests that the Design section should follow the objectives and I feel that conforming to this would improve the clarity of the paper considerably. Page 8 line 29 provides a statement of the design, however there should be more of an explanation of the key points in a multicenter, longitudinal, prospective, cluster-randomized clinical trial, for example why a cluster design was chosen and what are the main elements. Further, there is a design phase (objective 1) this requires description.

- We have added a more detailed explanation as to why we have chosen a cluster-randomized clinical trial for the effectiveness study.

The study setting eligibility criteria and participants are described in rather vague terms. This is particularly important in a cluster trial where clarity about issues such as the unit of randomisation and levels of participants for example- maternity clinic, midwife, pregnant woman, is essential. I presume that the primary care centres are the clusters, is consent to participation of the cluster obtained from the head of the Primary Care Centres.

- Clinical Effectiveness:

Yes, the primary health care centers are the clusters. The study population, all OSAKIDETZA-Basque Health Service public primary care centers that have midwives interested in optimizing their clinical practice by adopting innovations and have signed a collaboration agreement will be eligible, but this is not a consent. [All the public hospitals and primary care centers of the Basque Region are under Osakidetza, that is structured into several Integrated Health Organizations (IHO) spread throughout the Basque country. These integrated Health Organizations are groups of health centers within the same geographic area, with similar characteristics and population, under the direction of the same hospital]. The eligible centers will be randomized centrally, blindly, and electronically, in Biocruces-Bizkaia Research Institute, stratified by Integrated Healthcare Organization (IHO) and in a 1: 1 ratio. Centers of intervention will receive EMAeHealth and control centers not. What we are going to measure is the clinical effectiveness of the EMAeHealth intervention in improving quality of life and other patient variables, in the study of clinical effectiveness. Because individual allocation did not seem adequate, we randomized them by center. The informed consent is signed by the patients. We have added a more detailed explanation regarding the clinical profile of eligible patients. We agree in the relevance, in a cluster trial, about issues such as the unit of randomization- that is indicate- and levels of participants (for example- maternity clinic, midwife, pregnant woman) that is taken into account in the statistical methods we use for the analysis.

- Mixed-methods study, some indicators are measured at the center level, such as adoption and frequency of use.

Outcomes

The chosen outcomes seem relevant and interesting but they require more detailed description and justification for their use in this study.

- We have added a more detailed explanation about why we have chosen quality of life as primary outcome and about the instrument SF-36.

Indicators of the implementation process. This is important but under described. Why have focus groups been chosen rather than individual interviews? Will focus groups be undertaken in control sites – it might be important to know more about women's information seeking and access to maternal education, including commercially available Apps. A semi-structured script will be developed based on the theoretical constructs of the CFIR - what are these?

- The team has considered that the information it needs to collect is not of an intimate nature, in which case we would have opted for individual interviews. The focus group facilitates participant interaction, which in turn encourages the emergence of more themes, which is of interest to us. It is also less costly to conduct than individual interviews. As for the CFIR constructs, they are useful for evaluating an implementation. We have included a bibliographic citation for the reader who wants to learn more about the subject: Consolidated Framework for Implementation Research [28]
- Regarding the issue of including the control group in the focus group study, we have considered only to be conducted at the intervention sites because what we want to know is whether the intervention has been implemented correctly. The question of women's information seeking and access to maternal education, including commercial applications, although certainly very interesting, is unfortunately outside the objective of the study.

The participant flowchart is helpful.

- Thank you.

Study size, I am not able to assess the information provided here. I would be interested to see some more information that acknowledges that this is a cluster trial and the implications of that both for the calculation of the sample size and for the subsequent analysis of data.

- The analysis will be made with an intention-to-treat approach using mixed models (linear or generalized as appropriate to the variable under study), that take into account the hierarchical structure of the data (patients grouped by midwives, midwives grouped by center). We add that "The calculation took into account the cluster design of the study and it was carried out using the National

Institute of Health GRT sample size calculator* which assumes that the analysis will employ mixed-model regression methods”.

Page 13 Patient involvement. This section describes a previous study where a multidisciplinary group prioritised needs for Maternal Education. I think this may be what was alluded to in the introduction. If so, it would be helpful to have this work more fully described in the introduction.

- In the introduction, the background to this study has been somewhat better explained. In addition, we have rewritten this section because we believe it could be improved.

Strengths and limitations

The first three are statements which one may agree with but are not direct strengths and limitations of this study

- We have changed this section almost completely. It is now more focused on methodology, as suggested by the editors, and on strengths and weaknesses.

Thank you very much for all your comments. We believe they have helped to improve the quality of the protocol considerably. Best regards.

Reviewer: 2

Dr. Neeltje Crombag

Comments to the Author:

Thank you for giving me the opportunity to review your interesting study protocol. The topic and application is relevant and I would look forward to see the outcomes of this project. Below some suggestions to maybe improve your study protocol

- You are welcome. We are pleased to know that the topic has been of interest to you and we are sure that your contributions will undoubtedly be very useful to improve the protocol.

Strength and limitations

Here I would suggest rephrasing the sentences in which it becomes truly clear where the strengths and limitations are and why. For example:

EHealth can help the user of health services to take a more active role in decision-making and serve the professional to guide the patient in this process. It is a (major) strength of this study to include patients in the design and evaluation of these new technologies as this.....etc

- Both editors and reviewers have agreed on this point, so we have changed them almost completely.

Since the editors tell us that this section "should refer specifically to the methods of the reported study" and in accordance with their indications and those of reviewer 1, they have been left as is:

- o EMAeHealth is the prototype of a digital tool that supports decision making during pregnancy and postpartum. One strength is that it has been designed through Collaborative Action Research and is intended to be a complement to Maternal Education.

- o A hybrid design of efficacy trial and implementation will allow to evaluate the efficacy of

- EMAeHealth while collecting data on the implementation of the tool in "real world" conditions.

- o Both patients and professionals have participated in the previous Collaborative Action Research and will be involved in the evaluation of the tool and in its implementation.

- o A major strength is that the tool will be effective, will be adapted to our implementation context of application, and will be widely used because of the way it has been developed and evaluated;

- o A weakness is that the results might not be directly extrapolable to a different context and it would be necessary to adapt the tool to the context where it will be implemented.

Introduction:

First sentence should be split in 2 or 3 sentences

- Done

Methods and analysis:

It would be helpful if you could be a bit more detailed on the design of the tool. How is it designed, by which method (which design method is used) and by whom (are patients and healthcare professionals involved)?

- We sincerely believe that you are absolutely right in this. In writing the protocol, we made an error in a bibliographic citation (21), which we believe has conditioned both the understanding of the context

of the study and the stage we are at. This article 21 describes the steps taken to design the EMAeHealth tool. It reports that, prior to the study referred to in this protocol, extensive research was carried out on the needs of women in terms of Maternal Education, from the point of view of the women themselves and professionals, and an analysis of the evidence on needs, models of MS, sources of information most used on the Internet by women, etc. We have modified this bibliographical citation and clarified the first part of the introduction.

Also can you specify the inclusion criteria related to their clinical profile for eligible patients? Are they all low-risk at the beginning of pregnancy (assuming this as under midwifery care) but might be good to specify this.

- Yes. Spanish-speaking pregnant women who attend their first consultation with the midwife between 6 and 10 weeks of gestation. In terms of their clinical profile, they are low-risk pregnant women. We have added this information.

If I understand correctly, the tool is located randomly to a centre: so if centre A is allocated, centre A will use the ehealth, while centre B would not. But how can you be sure there are no pre-existing differences in the (quality of) care provided between centres? Or can you indicate why they are comparable? Also do all these centres serve the same population? Sometimes there are differences between because of location or the population they serve.

- “The eligible centers will be randomized centrally, blindly, and electronically, in Biocruces-Bizkaia Research Institute, stratified by Integrated Healthcare Organization (IHO) and in a 1: 1 ratio. “ We have added, within this paragraph of "Random assignment of centers" a description of what an IHO is:

“Integrated Health Organizations are groups of health centers within the same geographic area, with similar characteristics and population, under the direction of the same hospital. Therefore, we assume that there are no pre-existing differences in the quality of care provided between centers that belong to the same integrated health organization.”

We believe that the above description sheds some light on the question of comparability of the centers, at least for randomization. We are aware that there may be other variables to take into account, e.g. it is not possible for the intervention to be blind, but we will control for these in the analyses.

We greatly appreciate your contribution to improve the protocol. It has been very useful to us. Thank you very much. Best regards,

VERSION 2 – REVIEW

REVIEWER	Helen Cheyne University of Stirling, NMAHP Research Unit
REVIEW RETURNED	24-Apr-2022

GENERAL COMMENTS	Thank you for asking me to review this article. I have found following the rationale and study description difficult and I wonder if the main problem is that it requires further work to improve the quality of English throughout. I was also unable to locate the figures (1-3). Strengths and limitations – 4th bullet point needs re-written The rationale for the project is not very clear. Other work has been done but the reader should not have to read several other papers to understand this one. For example, what does this sentence mean (of ME) - However, it needs to be improved by reorienting the objectives, format, duration and content, to the needs of women in the current context. What is the evidence that current ME needs to be improved - there will be lots of really good education programmes – many now fully
---

	online. Is the main driver the pandemic? Is there a particular problem that needs to be overcome? Even if there is not that does not mean there is not room for another programme however, the authors need to say why, when making such claims. Paragraph 2 – starting The Research Team, could be clearer – what are the needs of women? What theoretical models? Maybe provide a brief summary of the key findings of the focus groups? This sentence - The evaluation of the effectiveness of digital tools aimed at health education during pregnancy is scarce and has had controversial results [22,23]; consequently, its realization becomes absolutely essential. Has been reworded but is still really unclear – maybe giving an example of the controversial results would help – also I don't know what you mean in saying 'its realization becomes absolutely essential' do you mean that it is important to evaluate the effectiveness of digital tools? Overall I don't think that the methods section is well described. For example At the Aim section no aim is provided – state an aim then the objectives. Given this is a protocol for a CRT the aim should take the form of a PICO or other structure that states the intervention, control and population - currently it is vague Outcomes What is the Spanish version of Lowe's CBSEI scales. Use the full title the first time of use What is - The concept of QV – what is QV? The SF 36 is explained but not the CBSEI or breast feeding self-efficacy scales And what about the clinical outcomes – bishops score etc What is the CFIR – and what are its domains.
--	--

REVIEWER	Neeltje Crombag
REVIEW RETURNED	12-Apr-2022

GENERAL COMMENTS	I believe the authors have addressed the issues raised in a constructive way, and in particular responded constructively to my concerns. The manuscript has improved significantly.
---

VERSION 2 – AUTHOR RESPONSE

Reviewer: 1

Dr. Helen Cheyne, University of Stirling Comments to the Author:

R1: Thank you for asking me to review this article. I have found following the rationale and study description difficult and I wonder if the main problem is that it requires further work to improve the quality of English throughout.

M: You are very welcome. Many thanks to you for the review. A professional native English translator with many years of experience in the translation of scientific texts has translated the submitted protocol. We have been working together for years and have had no previous problems with the quality of the English. Please, perhaps if you could be more specific about what is not understood, we would be pleased to rewrite it.

R1: I was also unable to locate the figures (1-3).

Figures 1 to 3 make it much easier to understand the protocol and study design, as they are 1) the description of the EMAeHealth design, 2) the clinical effectiveness study flowchart and 3) the Spirit timeline. We do not really understand why these figures are not accessible to the reviewer, as they have been uploaded to the application at all times. In the previous review, reviewer 1 even commented, in reference to figure 2, "The participant flowchart is useful". Please let the journal know if you still do not have access to the figures so that they can be made available to you.

R1: Strengths and limitations – 4th bullet point needs re- written.

We have rewritten point four.

R1: The rationale for the project is not very clear. Other work has been done but the reader should not have to read several other papers to understand this one.

We have made a summary of the previous works so that the need for the present study is better understood.

(1) For example, what does this sentence mean (of ME) – “ However, it needs to be improved by reorienting the objectives, format, duration and content, to the needs of women in the current context.
(2) “What is the evidence that current ME needs to be improved - there will be lots of really good education programmes – many now fully online. Is the main driver the pandemic? Is there a particular problem that needs to be overcome ? Even if there is not that does not mean there is not room for another programme however, the authors need to say why, when making such claims.

(1) You are right. Perhaps the phrase is too blunt. We believe we have now better justified this statement.

(2) We have explained now that “ the benefits of these programs to participants and their newborn infants remain unclear, and classes typically have not based on the expressed needs of attendees, but rather on the messages that the educators themselves believe they should impart”. This statement is supported by a Cochrane review of “Individual or group antenatal education programs”, two other reviews (a NICE review on the evidence for Antenatal Education and a McMillan review on the same topic), as well as our own work. According to our work, women, at least those in our setting, do not achieve much benefit from ME and, moreover, it is not tailored to their needs and preferences. Since ME is an intervention that comes from the health system, it should achieve some kind of benefit; consequently, if ME fails to do so, it should be improved.

(3) Paragraph 2 – starting The Research Team, could be clearer – what are the needs of women? What theoretical models? Maybe provide a brief summary of the key findings of the focus groups?
Done

(4) This sentence - The evaluation of the effectiveness of digital tools aimed at health education during pregnancy is scarce and has had controversial results [22,23]; consequently, its realization becomes absolutely essential. Has been reworded but is still really unclear – maybe giving an example of the controversial results would help – also I don't know what you mean in saying 'its realization becomes absolutely essential' do you mean that it is important to evaluate the effectiveness of digital tools?

We have replaced that sentence with the following: The evaluation of the effectiveness of digital tools aimed at health education during pregnancy is scarce and has had controversial results [22,23]: the quality, reliability and effectiveness of current pregnancy apps is yet to be determined [15,22,23], consequently, such studies to evaluate effectiveness are absolutely essential, even more so if the digital tool comes from the healthcare system.

R1: Overall I don't think that the methods section is well described. For example At the Aim section no aim is provided – state an aim then the objectives. Given this is a protocol for a CRT the aim should take the form of a PICO or other structure that states the intervention, control and population

Sorry, but since this is not the protocol of an RCT, but a protocol of a hybrid implementation-effectiveness study (composed of an RCT and a mixed-methods study), it is not possible to adapt it to the structure of a PICO. However, we have rephrased the main objective so that the reader can understand the dual focus of the study and we have also clarified some issues throughout the methodology.

R1: Currently it is vague Outcomes What is the Spanish version of Lowe's CBSEI scales. Use the full title the first time of use What is - The concept of QV – what is QV?

The SF 36 is explained but not the CBSEI or breast feeding self- efficacy scales And what about the clinical outcomes – bishops score etc What is the CFIR – and what are its domains.

(1) In relation to the self-efficacy scales for childbirth (CBSI-spanish version) and breastfeeding (BSES-SF spanish version), we have added the following information in the section of Outcomes:

“Self-efficacy is one of the key factors influencing women's confidence and ability to cope with childbirth and the Childbirth Self-Efficacy Inventory has been shown to be a valid and reliable instrument to measure maternal confidence in childbirth. The Spanish version of the Childbirth Self-Efficacy Inventory shows adequate psychometric properties (i.e. internal consistency and validity). Moreover, self-efficacy is a variable sensitive to external modifications, so we believe it may be suitable as one of the secondary measures to assess the effectiveness of using EMaHealth.

Similarly, breastfeeding self-efficacy is a mother's confidence in her ability to breastfeed and is highly predictive of breastfeeding behaviors. The Spanish version of the BSES-SF can be considered a valid and reliable measure of maternal breastfeeding self-efficacy.”

(2) Regarding the concept of QV: I'm sorry for this. It was a typographical error. Instead of QV it should be QoL (Quality of Life): "Quality of life (QoL) is a parameter that has been defined and recently taken into account as a health indicator by WHO [33]. The concept of QoL could be defined as an individual's assessment..."

(3) Regarding CFIR domains: In relation to the areas of the theoretical framework of the CFIR, we have added some information so that the reader knows what their purpose is and also what they consist of. However, as you are well aware, the space available for the protocol does not allow us to go into as much depth as we would like on each of the variables, theoretical frameworks... etc.

"A semi-structured script will be developed for the sessions, based on the theoretical constructs of the CFIR [28], which will also serve as the basis for the analysis. The CFIR includes 37 constructs within 5 main domains: intervention characteristics, outer setting, inner setting, individual characteristics and implementation process. Overall, the CFIR aims to help identify potential factors (i.e. barriers and facilitators) that are believed to influence implementation. Sessions will be audio-recorded and moderated by an expert in qualitative techniques while an assistant will take notes. Later the sessions will be transcribed and analysed".

Thank you very much for all your contributions. We are sure that the protocol has been substantially improved thanks to them. We hope that we have adequately resolved all the issues raised and also that we will be approved for publication in this prestigious journal.

VERSION 3 – REVIEW

REVIEWER	Helen Cheyne University of Stirling, NMAHP Research Unit
REVIEW RETURNED	16-Jun-2022

GENERAL COMMENTS	Thank you for addressing my previous comments. I have only a couple of very minor comments that require minor edits. The heading 'AIM' is included but no aim is stated. The objectives are provided and I think they are now clear. I suggest changing the heading to 'Objectives' or adding an aim. Abbreviations should be described in full on first use e.g CFIR should be Consolidated Framework for Implementation. The manuscript reads well but would benefit from a final edit to improve use of language.
--

VERSION 3 – AUTHOR RESPONSE

Reviewer: 1

Dr. Helen Cheyne, University of Stirling Comments to the Author:

"Thank you for addressing my previous comments. I have only a couple of very minor comments that require minor edits. The heading 'AIM' is included but no aim is stated. The objectives are provided and I think they are now clear. I suggest changing the heading to 'Objectives' or adding an aim. Abbreviations should be described in full on first use e.g CFIR should be Consolidated Framework for

Implementation. The manuscript reads well but would benefit from a final edit to improve use of language”.

You are very welcome. Many thanks to you for the review.

- On the subject of **aim and objectives**, we have finally changed the heading "AIM" to "Objectives" and added this at the end of the introduction: *“Therefore, we conducted this study to find out if complementing traditional perinatal care with a tool - designed through a collaborative action-research process - translates into better health outcomes for women than those obtained with traditional care alone. In addition, we want to know if this type of tool is useful and acceptable to its users in real-life conditions”*.
- **“Abbreviations** should be described in full on first use e.g CFIR should be Consolidated Framework for Implementation”: Done. Thank you very much.
- **“The manuscript reads well but would benefit from a final edit to improve use of language”**: Thank you; as you suggested, we have made a final revision to improve the use of English and make the text more coherent.